# Green Care: A Review of the Benefits and Potential of Animal-Assisted Care Farming Globally and in Rural America

**DOI:** 10.3390/ani7040031

**Published:** 2017-04-13

**Authors:** Brianna Artz, Doris Bitler Davis

**Affiliations:** Psychology Department, Mail Stop 3F5, George Mason University, 4400 University Drive, Fairfax, VA 22030, USA; bartz@gmu.edu

**Keywords:** green care, care farming, therapeutic horticulture, therapeutic communities, community care, rural America

## Abstract

**Simple Summary:**

The term Green Care encompasses a number of therapeutic strategies that can include farm-animal-assisted therapy, horticultural therapy, and general, farm-based therapy. This review article provides an overview of how Green Care has been used as part of the therapeutic plan for a variety of psychological disorders and related physical disabilities in children, adolescents and adults. While many countries have embraced Green Care, and research-based evidence supports its efficacy in a variety of therapeutic models, it has not yet gained widespread popularity in the United States. We suggest that Green Care could prove to be an effective approach to providing mental health care in the U.S., particularly in rural areas that are typically underserved by more traditional mental health facilities, but have an abundance of farms, livestock, and green spaces where care might be effectively provided.

**Abstract:**

The term Green Care includes therapeutic, social or educational interventions involving farming; farm animals; gardening or general contact with nature. Although Green Care can occur in any setting in which there is interaction with plants or animals, this review focuses on therapeutic practices occurring on farms. The efficacy of care farming is discussed and the broad utilization of care farming and farm care communities in Europe is reviewed. Though evidence from care farms in the United States is included in this review, the empirical evidence which could determine its efficacy is lacking. For example, the empirical evidence supporting or refuting the efficacy of therapeutic horseback riding in adults is minimal, while there is little non-equine care farming literature with children. The health care systems in Europe are also much different than those in the United States. In order for insurance companies to cover Green Care techniques in the United States, extensive research is necessary. This paper proposes community-based ways that Green Care methods can be utilized without insurance in the United States. Though Green Care can certainly be provided in urban areas, this paper focuses on ways rural areas can utilize existing farms to benefit the mental and physical health of their communities.

## 1. Purpose: The Promise of Green Care

The purpose of this review article is to provide an overview of the various types of farm-based interventions currently in use, and examine their efficacy in Europe and potential for use in the United States.

At the broadest level, Green Care is a term used to describe psychological, educational, social, or physical interventions that involve plants and/or animals (Haubenhofer, Elings, Hassink, and Hine, 2010) [1]. These types of interventions might be either passive (e.g., simply sitting in a natural setting) or active (e.g., caring for animals or sowing seeds and harvesting plants) (Sempik, Hine and Wilcox, 2010) [2]. Practices can occur in a variety of natural or naturalistic places, including parks, forests, gardens, grounds surrounding institutions, and farms (Hine, Peacock and Pretty, 2008 [3]). The common emphasis is the provision of services in a natural setting. This connection with nature is something that people seem to value, but it is often lacking in our increasingly urbanized world. While Green Care can include strategies such as ecotherapy, eco-education, therapeutic horticulture, wilderness, and nature therapy (Burls, 2008 [4]; Hine et al., 2008 [3]), the specific foci of this paper are animal-assisted care farming and therapeutic practices in farm settings (for an excellent review of all types of Green Care, see Sempik et al., 2010 [2]. In this limited context, facilities can range from those that are heavily reliant on farm production for income, to client-centered locations that emphasize care with the help of farm-like interventions. Farms built specifically for the benefit of clients may be an extension of an urban institution, rather than in a rural setting. More traditional, rural farms be less structured and allow more client independence. Care farming activities typically include crop farming, machinery use, or caring for livestock, with the goal of promoting physical, psychological, or social well-being. However, the specifics may vary based on the facility housing them (Hine et al., 2008 [3]).

A secondary purpose of this paper is to highlight a community-based intervention called care farming, which has been widely implemented outside the United States to provide care for a broad range of people, from those experiencing every-day stress, to those with severe medical or psychological disorders or disabilities. Specific populations targeted for this type of intervention may include at-risk youth, children with autism spectrum disorders, and the elderly (Types of Care Farms, 2015 [5]), in addition to those with psychological or physical challenges. In care farming, community-based farms that raise animals and/or crops partner with care providers to provide a healing space for individuals who choose to participate (Haubenhofer et al., 2010 [1]). Wilcox (2009) [6] visited a number of such farms across the European Union, and provides an overview of the unique forms, practices, and challenges of care farming. If a feasible model can be identified and is supported by research, there is the potential for implementing this type of intervention in American rural communities where other kinds of health care interventions may not be easily accessible.

This paper will focus on the efficacy of Green Care and, more specifically, care farming in the European Union and seek to determine its potential as a widely-used intervention in the United States, with a focus on rural areas in the U.S. Due to certain health-care-related challenges some may face due to factors such as geographical location, lack of access to health insurance, or socioeconomic status, care farming could be an effective alternative to more expensive, traditional interventions. This review will seek to evaluate the literature presented on the effectiveness of these interventions on a variety of mental, physical, educational and social problems. It will also seek to provide future directions for care farming in the United States, and how the type of strategies used in European practices might be translated to fit into the health care system of the United States.

## 2. Literature Review

Articles selected for inclusion in this review were largely from peer-reviewed, scientific journals with some online resources added to capture the full breadth of this topic. Data bases searched include the following: APA PsycNET, Psychology and Behavioral Sciences Collection, and National Center for Health Statistics. Google Scholar was used to search for articles outside of these databases. The citations from selected articles were also reviewed, and included if relevant. Search terms included the following: green care, farm care, equine-assisted therapy, farm-assisted therapy, therapeutic horticulture, and animal-assisted therapy. The articles included provided experimental or review-based evidence pertaining to Green Care and care farming, or spoke to the history of the practice of Green Care and involved farm-based interventions.

## 3. Equine-Assisted Therapy, Farm Animal-Assisted Interventions, and Therapeutic Horticulture

Though Green Care is a broad term essentially used to describe any intervention involving animals or nature, there are specific subsets of this term that will be the foci of this paper. Equine-assisted therapy, sometimes termed hippotherapy (or many other names) is the most broadly researched, and is often utilized with children and adolescents. Farm animal-assisted therapy can include caring or and interacting with a number of animals including cows, goats, chickens, etc. in order to provide a therapeutic benefit to an individual. Therapeutic horticulture involves caring for plant-life as an intervention strategy. These three techniques will be discussed in terms of their respective roles in assisting with psychological, educational, and physical problems. Thus, a brief overview of the three methods will be reviewed here. It is important to note that the three categories discussed below can, and often do, overlap during therapeutic interventions.

### 3.1. Equine-Assisted Therapy

Therapy methods involving horses have emerged as important for developing healthier emotional, social and behavioral functioning in children and adolescents. While sometimes considered a subset of farm animal-assisted therapy, the perception of horses as companion animals, rather than food animals, argues for their inclusion as a separate category. Within this subset of Green Care interventions, there are four different methods that can be implemented by therapeutic teams: (1) equine-assisted psychotherapy (EAP); (2) equine-assisted learning (EAL); (3) equine-facilitated psychotherapy (EFP); (4) equine-facilitated learning (EFL). All four methods utilize horses as co-therapists in different ways and with different goals for their specific therapeutic protocols. (Lee et al, 2016) [4] Since the therapeutic use of horses has been much more extensively categorized and researched then say, goats, this paper will refer to equine-assisted interventions as differing from farm animal-assisted interventions, though they may fall under the same category.

The Equine Assisted Growth and Learning Association (EAGALA) developed two therapeutic methods involving horses. Equine-assisted psychotherapy (EAP) includes an individual trained to work with horses and a mental health practitioner, such as a social worker or psychologist. This type of therapy utilizes the horse as a co-therapist to address certain desirable therapeutic goals such as confidence-building or self-efficacy, without mounting the horse. Equine-assisted learning (EAL) utilizes the same team members and similar unmounted activities; however, the goal is learning related rather than therapeutic, such as learning how to clearly express feelings or desires to family or friends. These therapeutic methods are often used together, incorporating both learning and therapeutic goals such as self-expression and communication skills. (Lee et al., 2016) [4].

Path International developed two techniques that are similar, but involve the additional element of mounting the horse as part of the therapeutic methodology. Equine-facilitated learning (EFL) does not require a mental health professional, since the goals are learning-related. However, since riding and related activities like jumping are usually included in this method, a team-member does need to have qualifications related to teaching horseback riding. Equine-facilitated psychotherapy (EFP) is similar to EAP in that a psychotherapist is involved and works with the client towards therapeutic goals; however, mounted activities are also included. (Lee et al., 2016) [7].

### 3.2. Farm Animal-Assisted Interventions

In this review, interventions involving farm-animals will be discussed, and are often combined with agriculture care. They also may overlap with care farming since care farming is considered the use of commercial farms, where animals may dwell (Haubenhofer et al., 2010) [1]. Farm animal-assisted therapies involve interacting with and caring for farm animals in order to promote well-being for those with mental, physical or educational problems. Therapies involving interacting with and caring for animals have been demonstrated to improve depression and anxiety, self-efficacy and other aspects of health in both in adults and children (Mallon, 1994 [8]; Pedersen, Martinsen, Berget, and Braastad, 2012 [9]; Pedersen, Nordaunet, Martinsen, Berget, and Braastad, 2011 [10]; Scholl et al., 2008 [11]). Animal-Assisted Interventions and therapies on farms can either have a very specific goal or use for the animal when interacting with the client and therapist, or the activities can be more general and client-based (Haubenhofer et al., 2010) [1]. As discussed previously, interventions involving horses can often be complicated and intricately planned, while interventions utilized with goats may follow a less strict path as in a study discussed later in the review (Scholl et al., 2008) [11]. Whether the interactions with farm animals are implemented in the form of a program or spontaneous and organic, the main goals of these interactions is to promote certain therapeutic, educational, motivational, or even social needs (Haubenhofer et al., 2010) [1].

### 3.3. Therapeutic Horticulture

Again, like many terms used to describe specific Green Care interventions, therapeutic horticulture can be utilized in tandem with the previous two methods. Therapeutic horticulture, or horticultural therapy, uses plants as a tool to help patients regain or learn skills important for normal functioning. This type of therapy can be thought of in terms of cognitive or occupational therapy, due to its goals of improving memory, balance and coordination, attention, and problem-solving. The stress-reduction, relaxation, and non-threatening aura of interaction with plant-life, combined with the mental and physical benefits of this type of therapy, are why is has been implemented successfully (Haubenhofer et al., 2010) [1].

A subset of therapeutic horticulture, social and therapeutic horticulture (STH), is methodologically much different from its non-social counterpart. Rather than specific goals of enhancing cognitive or physical processes, this branch of horticultural therapy emphasizes an overall increase in general well-being. The goal of STH is to provide community-based groups where social interaction can combine with gardening activities in a formal environment. Social and therapeutic horticulture has a large presence in the United Kingdom, and 90% of individuals utilizing these services there have either mental health problems or learning difficulties (Haubenhofer et al., 2010) [1].

## 4. Benefits to Children and Adolescents

### 4.1. Children with Behavioral, Emotional or Educational Problems

Mallon (1994) [8] highlighted, as an introductory point, that the majority of studies involving animal-assisted therapies or interventions do not focus on farm animals, a fact that remains true over 20 years later. Mallon’s study focused on the interactions between children with significant academic, behavioral, or emotional problems living at a residential facility. A questionnaire was administered to the children to determine why and how often they voluntarily visited the farm at the residential treatment site. Twenty seven percent of the respondents visited the farm daily and 71% visited the farm at least once a week. Thirty nine percent of the respondents reported that they decided to go to the farm specifically because they wanted to care for or spend time with the animals. Eighty two percent indicated that visiting the farm made them feel happy and that they would often choose to go there when they were feeling mad or sad because it alleviated their negative emotional states. Forty nine percent of respondents viewed horseback riding as their favorite activity. This indicates that, although the farm as a whole might have helped to mitigate negative affect, riding the horses might have been the main source of this emotional response. This study set the tone for future research delving into the farm as a whole and its potential for healing.

Forget Me Not Farm is located in Santa Rosa, California, and provides lessons on empathy and compassion through interactions with animals and agriculture to child victims of trauma and abuse. The program relies on trained volunteers and a structured routine to help children regain a sense of connection to, and trust in, the world around them. Children are given the opportunity to interact with and learn from cows, llamas, goats, pigs, chickens, and donkeys, among many other animals, to learn that living things should be cared for and will care about you in return. With many success stories, and an easily replicable program, The Forget Me Not Farm could be duplicated elsewhere. (Rossiter, 2006) [12].

A review of equine-assisted therapy by Lee et al (2016) [7] provided a promising analysis of the potential benefits of these kinds of therapies. A case study in Switzerland conducted by Chardonnens (2009) [13] demonstrated that a combination of EAP and EFP improved the behavioral and mental disorders of an eight-year-old boy. Through the combination of working with a horse, and a team including a mental health professional, the child demonstrated increased self-esteem, an improvement in interpersonal physical and emotional communication, and a sense of self-efficacy and responsibility. Dell et al. (2011) [14] reported that teenagers in treatment for substance abuse felt “in the present moment”, calmed, and connected to the horses in a way that reduced anxiety about the future, and male adolescents felt the horses provided them with an outlet to express affection. In the United States, Trotter et al. (2008) [15] explored the efficacy of equine-assisted counseling with children and teenagers who were having difficulty with peers or academics in comparison to a traditional counseling group. The study used the Behavioural Assessment System for Children (BASC) self and parent rating scales to determine the difference in outcomes based on treatment methods. Children reported statistically significant improvements in five areas of the BASC after therapy involving horses, and only four areas of significant improvement in the traditional counseling condition. Parents reported significant improvement in 12 areas of the BASC after the equine-assisted intervention, while the traditional counseling group only yielded one significant change in parent reports.

### 4.2. Children with Autism Spectrum Disorder

Bass, Duchowny and Llabre (2009) [16] compared a group of 19 children with Autism Spectrum Disorder (5–10 years) who participated in therapeutic horseback riding to a group of 15 children (4–10 years) in a wait-listed control group with the same condition after a 12-week therapy or wait-list period. The researchers used The Social Responsiveness Scale both before and after the intervention in order to determine if equine assisted activities had an effect on the social responsiveness of their participants. During the therapeutic horseback riding intervention, participants were involved in both mounted and unmounted activities including grooming, learning about the different parts of the horse’s body, riding at a walk or trot, and either verbalizing or signing whether their horse was walking, trotting, or at a halt as it occurred. When compared with the wait-listed group after the 12-week period, many areas of social responsiveness, such as social motivation and directed attention, increased, and inattention and distractibility decreased. The children in the treatment group also had significant improvements in their ability to break away from a sedentary activity, which is important due to the favorability of routine and object-orientation indicative of Autism Spectrum Disorder. This study suggests that children with autism can become more comfortable with socialization and flexibility after just a 12-week intervention involving horses.

Other researchers compared equine involved therapeutic methods for Autism Spectrum Disorder with another methods of treatment that has been demonstrated to improve certain aspects of functioning. Lanning, Baier, Ivey-Hatz, Krenek, and Tubbs (2014) [17] compared the effects of equine assisted and a social circles group on the quality of life of children with Autism Spectrum Disorder. Twenty-four children (ages 4–15) were split up into the treatment group (EAA) or comparison group (social group), and participated in a 12-week intervention. The equine assisted intervention group participated in weekly sessions where grooming lessons, safety lessons (helmet use, respect of animals, etc.), and mounted activities occurred over an hour-long period. The social circles group also attended a 12-week long intervention, where participants interacted with peers and facilitators to work on social interactions such as empathy, eye contact, and taking turns. Their results indicated that based on several measures aimed at identifying quality of life, the EAA intervention did make significant improvements in the quality of life of participants. The EAA group also yielded more improvement in general behavior areas when compared to the social group, as well as significant differences in social, emotional and physical functioning at just 6-weeks, when these results in the social group were not attained until the end of the study. While general behavior improvement was the only quality of life area that was significantly more effected than the social circle group, this study still strongly represents the potential of EAA in treating Autism Spectrum Disorder. It provides the same quality of life benefits of the social group circle, with the added benefit of improved general behavior.

Ferwerda van-Zonneveld, Oosting, and Kijlstra (2012) [18] interviewed seven farmers at Dutch care farms serving children with Autism Spectrum Disorders to assess the various models under which they operated. While there were differences in the main purpose of the facilities (agriculture or intervention) and the specific interventions they used, this paper provides an important perspective: that of the care farmer. While such facilities may play an important role in the lives of young people with ASD, they can only be effective if the needs of the farmers are met. In this case, a common concern was lack of knowledge about ASD. Future efforts in this area should plan to include an education component to address this issue.

## 5. Benefits to Adults with Psychiatric Disorders

The beginnings of Green Care aimed at healing psychiatric conditions in adults dates back to 1889, when Sir Frederic Truby King began running the Seacliff Asylum in New Zealand. The institution focused on the role of farming and agriculture in the health of human beings, and was one of few institutes at the time to combine psychiatry and agriculture. King believed that nature and fresh air were essential to recovering mental faculties that may have been lost due to a poor diet, upbringing, or overall desire to control oneself. King did not agree with the many harsh medications and sedatives that many other physicians used at the time; instead, he believed that a well-balanced diet, autonomy, and time spent participating in outdoor activities or interacting with farm animals would benefit his patients more than anything else. Though his success measured by release-rate was poor (6.7%, lower than the 9% average in New Zealand), some individuals benefited greatly from their time at Seacliff, such as a man released after six months free from suicidal ideation. King’s prototype of green care paved the way for more modern practices such as animal-assisted interventions and social and therapeutic horticulture, which are now widely used outside the United States (Stock and Brickell, 2013) [19].

### 5.1. Adults with Severe Psychiatric Conditions/Inpatient Facilities

Interactions with farm animals have also been demonstrated to benefit adults participating in intensive residential or out-patient treatment centers. Berget, Ekeberg and Braastad (2008) [20] explored how working with farm animals could improve the daily lives of patients in a treatment center who have severe psychiatric disorders and who have not held a job in at least the last six months. The participants in their study included individuals with schizophrenia, personality disorders, or severe mood disorders. Though 19 of the original sample dropped out due to lack of interest, boredom, or other personal reasons, the 35 people who continued with the program showed significantly more interest in the intensity and exactness of their work by the end of the 12-week intervention. Individuals with mood disorders also gained a significant increase in generalized self-efficacy by the end of the intervention. Those who continued with the study spent most of their time making physical contact with the animals: feeding, milking, and cleaning their sheds. This study indicates that, in patients who have an interest in the welfare of animals or are fond of them, a therapy involving interacting with animals can increase certain aspects of behavior that are important in daily life, such as intensity and exactness in work, as well as feelings of self-efficacy.

Cerino, Cirulli, Chiarotti, and Seripa (2011) [21] analyzed the effectiveness of equine-facilitated psychotherapy in reducing both positive and negative symptoms of schizophrenia in patients residing in various cities in Italy. Each of the 24 participants were involved in 40 sessions over a two-year period, occurring each week. Each hour long session involved unmounted grooming activities, an individual riding session, followed by a group riding session with two or three other participants. Using the Brief Psychiatric Rating Scale (BPRS) and the Positive and Negative Syndrome Scale (PANSS) to ascertain symptom levels before and after the treatment period, the researchers found significant improvement in many areas of the illness. Behaviors such as affective flattening, reduction in emotional intensity and range, and disorganized think without delusion were significantly different (*p* < 0.0001) and clinically meaningful (more than a 20% change). Behaviors including disorganized behavior, hallucinations, and delusions were reduced significantly (*p* < 0.0006) but were not considered clinically meaningful because it did not reduce by at least 20% percent. When considering the BPRS and PANSS as a whole, this therapeutic riding intervention made a significant symptom improvement among their participants. Since the methodology included no more than simple mounted and unmounted interactions with horses, this data suggests that a more rigorous treatment plan involving horses could be extremely beneficial to individuals with schizophrenia.

Though most care-farming efforts involve a farm where clients can come work, or an extension of a residential facility, there are residential communities that completely embrace the therapeutic properties of holistic, nature-forward healing. Hickey (2008) [22] describes a community in the Galloway hills of South West Scotland, where Buddhist meditation practice and green care gardening are combined to enhance the well-being and mental health of the community. The Lothlorien community was designed to help those in recovery from psychosis, but can provide services to a wide range of individuals with mental health problems as long as they are willing and motivated to take responsibility for themselves and make a change. The Lothlorien accepts members from all over Britain and focuses on the therapeutic aspect of community and universal acceptance and respect.

Life in the Lothlorien community is very structured: residents eat breakfast, then do chores, have a meeting, and work in the garden for two hours in the morning and afternoon. This type of structure and routine is described as highly valued by one resident, due to the chaotic nature and instability of their life before coming to the care community. Relaxation and mindfulness training are used in combination with gardening to increase awareness of the present moment, rather than focusing on intrusive and damaging thought processes. One resident describes mindfulness as something that provides a positive mood, allowing the focus to clear unsettling thoughts. Though no quantitative research has been conducted on the efficacy of the Lothlorien community, members who have transitioned outside the community speak to their significant improvement and the community offers a model for others hoping to create similar therapeutic environments (Hickey, 2008) [22].

A mostly Amish town in Mesopotamia, Ohio is home to Hopewell, a residential therapeutic community catering to those with serious forms of mental illness. The main goal of the program is to increase the global functioning of their clients utilizing farm-based interventions. Clinicians and administrators work together to care for the 40 adults housed in the community, running a daily program that includes exercise, therapeutic groups and farm activities. The main areas of focus include socialization, the regulation of emotional expression, and creating a sense of community through interaction with gardening, animal care and the arts. Discharge data supports the community’s efficacy, as their treatment model has succeeded in helping their residents increase their global functioning upon release (Loue, Karges and Carlton, 2014) [23].

### 5.2. Less Severe Psychiatric Conditions/Outpatient Care

Pedersen, Nordaunet, Martinsen, Berget, and Braastad (2011) [10] analyzed the effectiveness of a farm-based intervention with nineteen participants whom identified as at least mildly depressed on the Beck Depression Inventory, while also analyzing differences in anxiety and self-efficacy levels. For these participants, 12 weeks of twice-a-week farm work, which involved spending time milking and moving the animals, was associated with a significant decrease in depression and anxiety. Participant scores on the Generalized Self-Efficacy (GSE) measure increased from 22.6 to 25.6. Group-level depression and anxiety scores decreased from 28.7 to 19.1 and the mean anxiety score decreased from 54.4 to 49.6. Though not all farm-related behaviors were correlated with a significant decrease in depression or anxiety, and some were actually correlated with an increase, the overall experience of working with farm animals had a positive impact on the mental health of participants.

Pedersen, Martinsen, Berget and Braastad (2012) [9] continued the same line of research with clinically depressed individuals using the same 12-week farm intervention, adding a randomized control group. Though there were no statistically significant differences between the experimental and control groups, there was a significant change in the experimental group between the time of recruitment and after 12-week of farm-based intervention with regard to depression and self-efficacy. Participants also maintained these gains at a 3-month follow up. This study indicates that farm-based interventions can have a positive effect on treating depression in some individuals and that, even after exiting this kind of care, they can still benefit from their experience.

When considering the United Kingdom as a whole, the care farming community ranges from farms in the city, to farms connected to charities or institutions, to independent, rural farms. Forty nine percent of the care farms in the UK are funded by charitable organizations, while only 33% are funded by government-provided client fees, with the remaining farms receiving funding from social services or other trusts. Hine et al. (2008) [3] administered a questionnaire to individuals at UK care farming facilities with respondents including ex-offenders, those recovering from drug abuse, homeless, or with mental health problems. The questionnaire used various measures to determine if self-esteem and various levels of mood were improved after spending time on the care farm. Their analyses indicated that self-esteem and vigor were significantly increased, with significant decreases in anger, confusion, depression, fatigue and tension. Those surveyed indicated they enjoyed spending time in the fresh air, with the animals, and that they felt satisfied and worthy when spending time working on the farm. Due to United Kingdom pressures on healthcare, education, and prison services, and recent difficulties in the agricultural market, researchers conclude that care farming can offer a unique opportunity to remedy both cultural issues. This community effort can benefit varying subsets of the community, and bring citizens together toward the common goal of success in health, education, and farm care (Hine et al., 2008) [3].

Other countries in Europe have also determined that green care through care farming is beneficial for those with addictions or other illnesses. Elings and Hassink (2008) [24] conducted focus groups with 42 individuals involved in care farming programs in the Netherlands to analyze the perceived efficacy of such interventions. Participants with addiction indicated that the farm offered them a routine, a rhythm, and something to take their mind off of their affliction. Individuals spoke of increased levels of self-respect and self-esteem, along with a feeling of overall equality due to their time on the farm. Perfectionists realized that plants growing in every direction can still be beautiful, and those with extreme self-doubt realized they could help produce a plant they deemed as amazing. The care farms in Finland provide participants with a sense of self, community, and pride, but don’t necessarily stimulate plans for the future. “(Care farms) can be a resting place that some would prefer not to leave (for the time being), but also one that others do not (or cannot) regard as the last stop” (Elings and Hassink, 2008, p. 321) [24].

### 5.3. Benefits to Persons with Psychiatric Disorders Accompanied by Physical Disability

Working with farm animals not only positively influences the lives of those with mental health issues, but can also provide a healing outlet for those with multiple disabilities. Scholl, et al. (2008) [11] explored the behavioral effects of interacting with socialized goats on deaf individuals with varying behavioral issues. The behavioral issues among the participants included depression, hyperactivity, aggression, anxiety, low self-esteem, and communication issues. Participants were given the chance to play with the goats within the group of participants to increase social interaction, feeding and brushing the goats in order to care for them, and interacting with them by stroking or physical contact. A participant who was depressed demonstrated increased happiness and responsibility after connecting and forming a close relationship with one of the goats. Participants who were fearful at first were able to overcome their initial fear and showed increased skills, mobility and concentration when interacting with the goats. However, in a separate dining-room condition used to measure if benefits spilled over into other aspects of life, there were no significant changes.

## 6. Benefits to Adults with Physical Illness or Disability

Individuals with multiple sclerosis (MS) often suffer from issues with balance or postural instability. Since horseback riding is an activity that requires both balance and stable posture, a pilot study was conducted in order to determine if hippotherapy could be an efficacious therapy for these individuals. A total of 15 participants (24–72 years) with MS, and balance deficits, were recruited to participate in a weekly therapeutic riding program for 14 weeks. The first nine people to respond to a recruitment flyer were assigned to the experimental group, but they could only find six participants willing to serve as a control group. The therapy sessions were a total of 40-min, with a 5-min warm-up and cool-down. The entire session was mounted, but not with riding instructions. An experienced handler would guide the horse, and the main goal of the participants was simply to remain balanced as the horse’s body moved. They were also instructed to change positions on the horse, sit sideways, backwards, and engage in other activities that challenged their postural control. Using the Berg Balance Scale (BBS) and the Tinetti Performance Oriented Mobility Assessment (POMA), pre and post intervention balance was assessed. While there was no significant improvements in either scale in the control group, the experimental group experienced significant improvements in balance and mobility indicated by both scales. There was even a significant difference in the final scores of the BBS and POMA when the experimental and control groups were compared. This pilot study shows promising results for treating balance issues caused by MS using hippotherapy, and could possibly extend to other physical disorders affecting balance. (Silkwood-Sherer, 2007 [25]).

## 7. Who Knows about Green Care?

It is evident that there are definite benefits from utilizing green care techniques to treat a variety of problems human beings may encounter throughout the span of their lives. It is evident that European researchers find this topic to be important and have extensively researched it; however, the only branch of green care heavily researched in the United States is hippotherapy. While there is a large body of research investigating equine assisted activities with children and adolescents, it is lacking with adult populations, especially involving potential mental health benefits. On the other hand, child interventions are heavily related to therapeutic horseback riding, while there are few studies examining how a farm as a whole could benefit youth. In order to fully understand the gaps in the literature, and the culturally differences in research interests between the United States and Europe, one much first know who knows about it and believes in its efficacy from an anecdotal level.

Berget, Ekeberg and Braastad (2008) [20] explored the attitudes of Norwegian therapists and farmers towards Green Care before beginning a study on animal-assisted therapy using farm animals. They found that among the 60 mental health professionals and 15 farmers surveyed, most respondents knew either some or a large amount about Green Care, but actual experience with Green Care was relatively low among both therapists and farmers. In terms of whether or not animals in general could contribute positively to traditional therapy, farmers thought the effect of animal-assisted interventions could be significantly larger than the therapists. Within the sample of therapists, females were much more likely to believe that animals could positively contribute to traditional therapy. Both farmers and therapists believed that farm animals could provide an additional therapeutic benefit that pets could not provide, but again, females were more positive regarding the additional benefits.

This study indicates that both therapeutic and farming professionals had at least some knowledge of Green Care and farm animal-assisted therapy in Norway before entering into the protocol, but that many do not have previous experience. Considering the prevalence of Green Care in Europe, it is unsurprising that those surveyed have knowledge of it. However, the level of knowledge that Americans have about Green Care has not been explored. Since the United States does not have government-sponsored health care programs, it is likely that any care farming efforts would be most effective if enacted at the community level, promoting a symbiotic relationship between clients and farmers. Therefore, replicating the work of Berget, et al. with a United States rural population could be advantageous in determining its potential at a broader level. If the rural American population does not even know about green care and its potential to build happier, healthier communities, it may never be utilized to its fullest potential.

## 8. Need for Community Mental Health Care in Rural America

Though there are certainly countless farms where green care or care farming could take place, it doesn’t seem to be happening at a rate that can keep up with the needs of rural communities. According to the American Psychological Association (The mental and behavioral health needs of rural communities [26]), 60% of people living in rural communities in the United States live in areas with a shortage of mental health professionals. They assert that, even if primary care physicians are available, they do not have the necessary training to adequately treat and diagnose potential problems, leading to ineffective treatments. Due to the high prevalence of mental health care shortages in rural areas, developing community care farming practices could be beneficial to the mental health of Americans living in rural communities. The programs developed could also aid in the overall production of resources from the farms, creating a symbiotic and reciprocal relationship between farmers and the volunteer clients who visit them.

46 million Americans, or 15%, live in rural areas where they are at a higher risk for heart disease, cancer, unintentional injuries, chronic lower respiratory disease, and stroke than the 85% of Americans in designated urban areas. Rural Americans have a higher poverty rate, and tend to live more sedentary lifestyles than urban Americans. On top of higher rates of obesity and cigarette smoking, those living in rural areas are also more likely to be without health insurance or adequate health care. The CDC recommends various steps in order to reduce the higher risk of death in these populations, including cancer and blood pressure screening, promoting a healthier lifestyle through food, exercise, and fewer prescriptions for opioid pain killers, and finally encouraging individuals to quit smoking. (Rural Americans at higher risk, 2017) [27].

While traditional, Western medicine certainly has the means to tackle these problems, it seems that they haven’t. Green Care efforts in rural America could be a community based effort to get individuals more active, growing produce in a community garden which they can take home to their families, promote increases in self-efficacy which would provide them the will-power to quit smoking cigarettes or using other substances, among many other outcomes. Community care farming could provide a new and exciting outlet for rural Americans who are not already engaged in farming as a profession, and create healthier rural communities across the United States.

## 9. Future Directions for Care Farming

### 9.1. Future Directions for American Care Farming

Due to the lack of mental health access for the majority of Americans living in rural communities, community care farming could promote healthier mental states for those who participate. Research has shown that working with horses and other farm animals, as well as working in a garden setting, can increase self-esteem, reduce depression and anxiety, and benefit the overall well-being of people (Lee et al., 2016 [7]; Hickey, 2008 [22]; Hine et al., 2008 [3]). Self-efficacy is also a benefit of certain care-farming techniques (Berget et al., 2008) [20]), which could provide rural Americans with more mental tools to combat personal choices such as a sedentary lifestyle, poor food choices, or smoking cigarettes or using other drugs. In fact, the unique qualities of care farming, including its emphasis on self-empowerment, personal strengths, and community involvement, make it applicable for a wide variety of populations, each of which may identify different benefits from the experience (Hassink, et al., 2010) [28].

In order for these kinds of interventions to begin, more research needs to be conducted in the United States. Though the evidence from European studies is promising, it may not be generalizable to the attitudes and belief-systems of rural Americans. University researchers could partner with local farmers, and invite participants to join their study under the premise of a community gathering. If funding was available, participation could be incentivized with compensation. This may be a particularly advantageous recruitment tool given the socio-economic climate of rural America.

Once there is sufficient evidence to support care farming efforts in the United States, social workers or health care providers in the community could partner with local farmers to implement therapy methods informed by the research. Individuals with unused land could be encouraged to donate a portion of it to a community garden where they, in return, could enjoy the harvests of the community gardeners. Individuality is a common theme in American culture, but if that barrier could be broken, community care farming could possibly help alleviate some of the issues which may arise simply because you dwell in rural America.

### 9.2. Future Directions for Care Farming, Universally

As mentioned previously, there is a large body of literature reporting on equine assisted interventions with children, but a large gap when it comes to the advantages of its use with adults. Future research could focus on this subset of green care, and explore its potential with older populations. Alternatively, there is little research with children that does not include horses. Therapeutic horticulture and farm animal-assisted therapies with animals other than horses may also be beneficial to children.

Additionally, the responsibility of the psychological community focused on green care is to disseminate the knowledge gained through previous research and continue to explore the benefits of Green Care and farm animal-assisted therapy. Without spreading the word, communities will never know about the healing places being cultivated in their own backyards.

### 9.3. Future Directions for Research

There are a number of promising directions for future research in this area that could inform further development of Green Care and care farming.

One such direction might involve comparisons of the provision, use, and benefits of care farming based on the divergent models already in existence. Di Iacovo and O’Connor’s 2009 [29] report on the SoFar Project (the acronym represents Social Farming, another term for care farming) identifies the different views and practices of care farming across the European Union. While all models include a multifunctional view of agriculture, they each are deeply rooted in the agricultural and cultural traditions of the regions in which they arose. The SoFar Project report provides ideas for many experiments comparing the different forms of care farming that have emerged naturally.

Research that specifically evaluates farm animal-assisted interventions might fruitfully build on the research that has been conducted on therapies involving companion animals. Kazdin (2017) [30] recently published an article with suggestions for moving forward in acquiring evidence for pet-assisted therapy that could be applied to farm care settings.

In thinking about the future of research on care farming, it is important that the perspectives of all stakeholders are included, including the care providers. Hassink, Hulsink, and Grin (2016) [31] interviewed farmers at ten different types of care farms in the Netherlands to identify the various entrepreneurial strategies employed to develop effective and profitable multi-use agricultural settings. In the United States, multi-use farms already exist outside of the therapeutic community, providing tours, activities, classes, and events. These might also be studied with a goal of better understanding the value of such strategies for the farmers, obstacles to entry, and benefits to the community.

Finally, it is important to note that Green Care and care farming are, by nature, multidisciplinary efforts that must be studied from a multidisciplinary perspective (Sempik, et al., 2010 [2]). In addition to the obvious input from fields such as psychology and agriculture, many other disciplines (e.g., sociology, cultural studies, economics, and business) might provide important perspectives on the development of these burgeoning strategies for providing therapeutic interventions.

## 10. Conclusions

Care farming has proven to be an effective form of alternative care in Europe and has benefitted individuals with mental and physical health issues. While farm-based interventions are being combined with more traditional therapeutic models in Europe, the United States has only just begun exploring their potential. Due to the efficacy of this type of intervention with European populations, one may conclude that care farming could be equally advantageous in the United States. However, further research is necessary to investigate the willingness of American populations to utilize this type of intervention, as well as to determine how it could be best implemented. European care farming efforts are often funded by the government, while health care coverage in the United States is the responsibility of the individual. Due to this disparity in health care access, it may be even more important for care farming to be adopted in the U.S., specifically in rural populations with limited access to health care. These differences between cultures must be explored before European empirical support for care farming can be applied to United States care farming efforts.

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
