# Peer review of "Green Care: A Review of the Benefits and Potential of Animal-Assisted Care Farming Globally and in Rural America"

_animals, 2017, doi:10.3390/ani7040031_

Round 1

Reviewer 1 Report

After reviewing this article I was very impressed about the value and potential of AA care farming. The reviewers believe that this option could have tremendous value for providing mental health support of many people and do a good job documenting the literature. That being said, I feel more attention could have been given to the actual findings of the project and the original suggestions that they could give the readers to move forward. How can we strengthen this area and perhaps make it a more viable resource?

Could the sample be increased to gain more insights. I believe that a larger population could lend more reliability to the comments and insights generated. It is also suggested that the authors work on the conclusions and provide ideas for mental health professionals to consider in utilizing this underdeveloped resource. Collaboration between the farming community and professionals in mental health would be a good beginning ( as we have seen in other countries).

Author Response

Reviewer 1 comments:

After reviewing this article I was very impressed about the value and potential of AA care farming.

The reviewers believe that this option could have tremendous value for providing mental health

support of many people and do a good job documenting the literature. That being said, I feel more

attention could have been given to the actual findings of the project and the original suggestions

that they could give the readers to move forward. How can we strengthen this area and perhaps

make it a more viable resource?

Due to concerns about the small sample size of our pilot survey, we decided to delete it from this manuscript.  Once we have more data and the project is complete, we plan to submit it as a separate article.

In terms of moving forward, we have expanded the section on future directions for green care and animal-assisted therapy (Section 9, Future Directions for Care Farming, page 11) to include a section specific to the U.S. (Section 9.1, Future Directions for American Care Farming, page 11, line 426) and another that addresses global issues (9.2, Future Directions for Care Farming, Universally, page 11, line 448).

Could the sample be increased to gain more insights. I believe that a larger population could lend

more reliability to the comments and insights generated. It is also suggested that the authors work

on the conclusions and provide ideas for mental health professionals to consider in utilizing this

underdeveloped resource. Collaboration between the farming community and professionals in

mental health would be a good beginning ( as we have seen in other countries).

Due to concerns about the small sample size of our pilot survey, we decided to delete it from this manuscript.  Once we have more data and the project is complete, we plan to submit it as a separate article.

With regard to possible collaborations between rural communities and mental health professionals, we have expanded the section on future directions for green care and animal-assisted therapy (Section 9, Future Directions for Care Farming, page 11) to include a section specific to the U.S. (Section 9.1, Future Directions for American Care Farming, page 11, line 426) and another that addresses global issues (9.1, Future Directions for Care Farming, Universally, page 11, line 448).

Interestingly enough, many of the green care farms about which we read originated because the spouse of a farmer was working in a therapeutic environment (e.g., as a social worker or psychologist) and saw the potential for using farms and/or animals as an enhancement to treatment.  So, these early collaborations evolved organically. 

Reviewer 2 Report

This paper attempts to discuss green care and the potential therapeutic role of interactions with animals in the context of these practices.

While the manuscript has some degree of novelty, the authors should not underestimate a great bulk of work testifying the therapeutic role of e.g. horses in animal-assisted interventions such as equine-assisted therapies. A number of recent publications are not taken into consideration, especially those referring to the efficacy with adults. The only mentioning of one review on EAT in the context of psychotherapy is not sufficiently informative.

One may want to cite:

              Cerino, Borgi et al. Riv. Psichiatria, 2016;

Bronson, C., Brewerton, K., Ong, J., Palanca, C., & Sullivan, S. J. (2010). Does hippotherapy improve balance in persons with multiple sclerosis: a systematic review. Eur J Phys Rehabil Med, 46, 347-353.

Cerino, S., Cirulli, F., Chiarotti, F., & Seripa, S. (2011). Non conventional psychiatric rehabilitation in schizophrenia using therapeutic riding: the FISE multicentre Pindar project. Ann Ist Super Sanita, 47(4), 409-414.

Freund, L. S., Brown, O. J., & Huff, P. R. (2011). Equine-assisted activities and therapy for individuals with physical and developmental disabilities: An overview of research findings and the types of research currently being conducted. In P. McCardle, S. McCune, J. A. Griffin, L. Esposito, & L. S. Freund (Eds.), Animals in our lives: Human animal interaction in family, community and therapeutic settings. Baltimore: Paul H. Brookes.

Lanning, B. A., Baier, M. E., Ivey-Hatz, J., Krenek, N., & Tubbs, J. D. (2014). Effects of equine assisted activities on autism spectrum disorder. J Autism Dev Disord, 44(8), 1897-1907.

An important caveat of the paper is that in the introduction, it is not discussed which are the basis for the potential role of human-animal interactions in human health and the authors should immediately touch upon this. The authors may want to cite: Cirulli, F., Borgi, M., Berry, A., Francia, N., & Alleva, E. (2011). Animal-assisted interventions as innovative tools for mental health. Ann Ist Super Sanita, 47(4), 341-348.

Author Response

Reviewer 2 comments:

This paper attempts to discuss green care and the potential therapeutic role of interactions with

animals in the context of these practices.

While the manuscript has some degree of novelty, the authors should not underestimate a great

bulk of work testifying the therapeutic role of e.g. horses in animal-assisted interventions such as

equine-assisted therapies. A number of recent publications are not taken into consideration,

especially those referring to the efficacy with adults. The only mentioning of one review on EAT in

the context of psychotherapy is not sufficiently informative.

The section on hippotherapy has been expanded (Section 3.1, Equine-Assisted Therapy, page 3, line 83).  Also added are sections on the use of other farm animals in therapy (Section 3.2, Farm Animal-Assisted Interventions, page 3, line 10) and horticultural therapy (Section 3.3, Therapeutic Horticulture, page 4, line 126).

Two sections specifically related to the treatment of adults have been added.  Section 5, Benefits to Adults with Psychiatric Disorders (page 6, line 215), includes subsections on inpatient treatment, outpatient treatment, and comorbid psychiatric and physical disabilities.  Section 6, Benefits to Adults with Physical Illness or Disability (page 9, line 347), specifically addresses physical disabilities.

One may want to cite:

Cerino, Borgi et al. Riv. Psichiatria, 2016;

Bronson, C., Brewerton, K., Ong, J., Palanca, C., & Sullivan, S. J. (2010). Does hippotherapy

improve balance in persons with multiple sclerosis: a systematic review. Eur J Phys Rehabil Med,

46, 347-353.

Cerino, S., Cirulli, F., Chiarotti, F., & Seripa, S. (2011). Non conventional psychiatric rehabilitation

in schizophrenia using therapeutic riding: the FISE multicentre Pindar project. Ann Ist Super Sanita,

47(4), 409-414.

Freund, L. S., Brown, O. J., & Huff, P. R. (2011). Equine-assisted activities and therapy for

individuals with physical and developmental disabilities: An overview of research findings and the

types of research currently being conducted. In P. McCardle, S. McCune, J. A. Griffin, L. Esposito,

& L. S. Freund (Eds.), Animals in our lives: Human animal interaction in family, community and

therapeutic settings. Baltimore: Paul H. Brookes.

Lanning, B. A., Baier, M. E., Ivey-Hatz, J., Krenek, N., & Tubbs, J. D. (2014). Effects of equine

assisted activities on autism spectrum disorder. J Autism Dev Disord, 44(8), 1897-1907.

Thank you for these suggestions!  The in-text citation for Cerino, Cirulli, Chiarotti, and Seripa (2001) can be found on page 6 (line 244).  The reference number is 17 and can be found on page 12, line 500.  The in-text citation for Lanning, Baier, Ivey-Hatz, Krenek, and Tubbs (2014) can be found on page 5, line 199.  The reference number is 14 and can be found on page 12, line 43.

An important caveat of the paper is that in the introduction, it is not discussed which are the basis for

the potential role of human-animal interactions in human health and the authors should immediately

touch upon this.

The authors may want to cite: Cirulli, F., Borgi, M., Berry, A., Francia, N., &

Alleva, E. (2011). Animal-assisted interventions as innovative tools for mental health. Ann Ist Super

Sanita, 47(4), 341-348.

Since the Cirulli, Borgi, Berry, Francia, and Alleva (2011) article has companion animals, specifically dogs, as its main focus, we do not consider it a good fit for this review, which is focused on farm animals.  It is not clear that the mechanisms or applications of therapies involving companion animals can be directly transferred to farm animals.  It seems that research specifically addressing this question could prove to be quite fruitful. 

Reviewer 3 Report

This could be an interesting paper. However in the present form it can not be published. In the summary and in part 1, the section purpose, it is indicated that the aim is providing an overview of the various types of farm-based interventions. This is not the case. The authors have selected a few studies ad randomly. There is no explanation why these studies were chosen. There has been no description of a literature review. Studies from the Netherlands and Italy are lacking. It seems that only studies dealing with participants with mental illness were selected. This is not explained however. There are several interesting care farming papers from Simone de Bruin, Francesco diIacovo, Sorana Iancua and Jan Hassink that have not been included. 

There is also no logical line in the paper. Why is chapter 2 dealing with farm based interventions and chapter 3 with equine assisted farm interventions and chapter  4 with green care. Chapter 3 seems a repetition of chapter 2. 

In Chapter 4 it is not clear why so much attention is given to the Lothlorien community. 

In Chapter 4 the study of Elings and Hassink is not situated in Finland but in the Netherlands. 

In Chapter 5 only two US studies have been incorporated. There are more care farms in the US and more papers. 

In Chapter 6 it is not clear why the focus is on rural America. Care farms can also be important for urban areas. 

Chapter 7 is about another rather ad hoc topic. Here only one study is incorporated. This is too limited. There are more Norwegian studies dealing with the same topic.

Chapter 8. This is a strange place for the materials and methods section.

Chapter 9. It is not clear who are the participants. It is not clear why only nine participants were included. The conclustions are based on only 9 respondents. We do not know what their background is. Why were respondents selected by a poultry website and facebook of a local veterinary office. Why was this village selected?  So it is not clear if it has any value. Question 3 of the questionnaire deals with mental illness.Was this the focus? It is not clear.    

Chapter 10. Again it seems that the paper focusses on mental illness. This has not been made clear in the introduction or the objectives of the paper. 

Chapter 11. There has been quite a lot of research dealing with the benefits of care farms for different populations (elderly, unemployed, mental illness, problem youth).

My conclusion is that the basis of this paper is very weak. The review on care farming is far from complete and lacks a focus. The methods section is very weak and the number of respondents for the questionnaire is too low. The purpose of the paper is not clear.

Author Response

Reviewer 3 comments:

This could be an interesting paper. However in the present form it can not be published. In the

summary and in part 1, the section purpose, it is indicated that the aim is providing an overview of

the various types of farm-based interventions. This is not the case. The authors have selected a few

studies ad randomly. There is no explanation why these studies were chosen. There has been no

description of a literature review. Studies from the Netherlands and Italy are lacking. It seems that

only studies dealing with participants with mental illness were selected. This is not explained

however. There are several interesting care farming papers from Simone de Bruin, Francesco

diIacovo, Sorana Iancua and Jan Hassink that have not been included.

A description of the literature review process has been added (Section 2, Literature Review, page 2, line 63).

A section that addresses farm-based therapies for children with behavioral and educational issues, as well as psychological disorders, has been added (Section 4.1, Children with Behavioral, Emotional or Educational Problems, page 4, line 143).  Two sections specifically related to the treatment of physical disabilities in adults have been added.  Section 5, Benefits to Adults with Psychiatric Disorders (page 6, line 215), includes subsections on inpatient treatment, outpatient treatment, and comorbid psychiatric and physical disabilities.  Section 6, Benefits to Adults with Physical Illness or Disability (page 9, line 347), specifically addresses physical disabilities.

Hassink’s work has been added.  The in-text citation for Haubenhofer, Elings, Hassink, and Hine (2010) can be found on page 1 (line 32).  The reference number is 1 and can be found on page 12, line 464.  The in-text citation for Elings and Hassink (2008) can be found on page 8 (lines 323 and 332).  The reference number is 20 and can be found on page 1, line 506. 

There is also no logical line in the paper. Why is chapter 2 dealing with farm based interventions

and chapter 3 with equine assisted farm interventions and chapter 4 with green care. Chapter 3

seems a repetition of chapter 2.

The entire manuscript has been reorganized as follows:

Original manuscript – highlighted text indicates sections that have been deleted, substantially revised, and/or incorporated into new subject headings for the sake of clarity and an improved flow of topics.

1.       Purpose: The Promise of Green Care and Care Farming

2.       Farm-Based Interventions: Considering the Farm as a Whole

3.       Equine-assisted Farm Interventions

4.       Green Care in Europe

5.       Green Care and Care Farming in the United States

6.       Need for Community Mental Health Care in Rural America

7.       Who Knows About Green Care?

8.       Materials and Methods *

9.       Results*

10.   Tables*

11.   Future Directions for American Care Farming

* Due to reasonable concerns about the small size of the sample in our pilot survey, we decided to remove it from this paper.  When we have completed that project (i.e., revised the survey and expanded our subject pool), we plan to submit it as a separate manuscript.

Revised manuscript – highlighted text indicates sections that have been added or substantially revised from their original form.

1.       Purpose: The Promise of Green Care and Care Farming

2.       Literature Review

3.       Equine-Assisted Therapy, Farm Animal-Assisted Interventions, and Therapeutic Horticulture

3.1 Equine-Assisted Therapy

3.2 Farm Animal-Assisted Interventions

3.3 Therapeutic Horticulture

4.       Benefits to Children and Adolescents

4.1 Children with Behavioral, Emotional or Educational Problems

4.2 Children with Autism Spectrum Disorder

5.       Benefits to Adults with Psychiatric Disorders

5.1 Adults with Severe Psychiatric Conditions / Inpatient Facilities

5.2 Less Severe Psychiatric Conditions / Outpatient Care

5.3 Benefits to Persons with Psychiatric Disorders accompanied by Physical Disability

6.       Benefits to Adults with Physical Illness or Disability

7.       Who Knows About Green Care?

8.       Need for Community Mental Health Care in Rural America

9.       Future Directions for Care Farming

9.1 Future Directions for American Care Farming

9.2 Future Directions for Care Farming, Universally

In Chapter 4 it is not clear why so much attention is given to the Lothlorien community.

In our opinion, Lothlorien is a community that has received wide-ranging attention and serves as an example.  A description of the Hopewell community has been added as an example from the U.S.  There are, of course, a number of these communities around the world, but attempting to describe all of them is outside the scope of this review.

In Chapter 4 the study of Elings and Hassink is not situated in Finland but in the Netherlands.

Corrected – thank you for catching that error!

In Chapter 5 only two US studies have been incorporated. There are more care farms in the US and

more papers.

Since detailed descriptions of the many care farms around the world is outside the scope of this review, we have limited ourselves to one example from outside the U.S. (Lothlorien) and one from the U.S. (Hopewell).

In Chapter 6 it is not clear why the focus is on rural America. Care farms can also be important for

urban areas.

The point we hope to make is that care farms might be particularly useful in rural regions with few mental-health-care options, but numerous green spaces.  Section 8, Need for Community Mental Health Care in Rural America, page 10, line 398, provides justification.

Chapter 7 is about another rather ad hoc topic. Here only one study is incorporated. This is too

limited. There are more Norwegian studies dealing with the same topic.

In this section, we hope to illustrate the need for further research on this topic in the U.S.

Chapter 8. This is a strange place for the materials and methods section.

Due to reasonable concerns about the small size of the sample in our pilot survey, we decided to remove it from this paper.  When we have completed that project (i.e., revised the survey and expanded our subject pool), we plan to submit it as a separate manuscript.

Chapter 9. It is not clear who are the participants. It is not clear why only nine participants were

included. The conclustions are based on only 9 respondents. We do not know what their

background is. Why were respondents selected by a poultry website and facebook of a local

veterinary office. Why was this village selected? So it is not clear if it has any value. Question 3 of

the questionnaire deals with mental illness.Was this the focus? It is not clear.

Due to reasonable concerns about the small size of the sample in our pilot survey, we decided to remove it from this paper.  When we have completed that project (i.e., revised the survey and expanded our subject pool), we plan to submit it as a separate manuscript.

Chapter 10. Again it seems that the paper focusses on mental illness. This has not been made clear

in the introduction or the objectives of the paper.

While the main focus of this manuscript is, indeed, the use of care farming as part of the treatment of psychological disorders, we have added sections on physical disabilities and comorbidity.  The revised manuscript includes a section that addresses farm-based therapies for children with behavioral and educational issues, as well as psychological disorders (Section 4.1, Children with Behavioral, Emotional or Educational Problems, page 4, line 143).  Two sections specifically related to the treatment of physical disabilities in adults have been added.  Section 5, Benefits to Adults with Psychiatric Disorders (page 6, line 215), includes subsections on inpatient treatment, outpatient treatment, and comorbid psychiatric and physical disabilities.  Section 6, Benefits to Adults with Physical Illness or Disability (page 9, line 347), specifically addresses physical disabilities.

Chapter 11. There has been quite a lot of research dealing with the benefits of care farms for

different populations (elderly, unemployed, mental illness, problem youth).

We believe that the use of care farming in the treatment of psychological disorders has been included in this manuscript.  Since our focus is on psychological disorders and, secondarily, related physical disabilities, social conditions like unemployment are outside the scope of this paper.  We did consider including care farms that provide housing for the elderly, but in the end determined that age alone does not necessarily coexist alongside disorders/disabilities, so decided against it.   

My conclusion is that the basis of this paper is very weak. The review on care farming is far from

complete and lacks a focus. The methods section is very weak and the number of respondents for

the questionnaire is too low. The purpose of the paper is not clear.

Thank you for your very helpful and detailed comments.

Round 2

Reviewer 1 Report

I feel that the AACF  should have been highlighted the most in this paper and integrating E.A.T. clouds the readers understanding. I feel that the AACF interventions are the most unique contributing aspect of the paper and they need to be elaborated.

What suggestions do you feel would help practitioners consider utilizing AACF in their therapeutic regime?  Going more into this and giving your insights on how to do apply these procedures would be extremely valuable.

Reviewer 2 Report

comments and suggestions have been mostly incorporated

Reviewer 3 Report

I think the paper has improved considerably. The authors have adopted the suggestions of the reviewers.  The think that the introduction (Purpose) can still be improved. The paper still does not give a good overview of the various types of care farming in Europe. 

Please add research findings related to the unique quality of care farms. It offers a combination of green, social qualities (see Hassink et al 2010 in Health and Place).

Please add that in Europe that the focus differs between countries; focus on integrating health care and agriculture like in the Netherlands and Norway and focus on integrating agricultura and labour (Italy and France). See the book of the Sofar project edited by DIiacovo.and O Connor.

Please add a reference about the diversity of care farms (agriculture oriented or care oriented) and challenges of farmers to combine agriculture and care. See the papers of Hassink et al. in Rural sociology, Journal of Rural Studies or Sociologia Ruralis.  

4.2 Please add the paper of Ferwerda about care farms published in NJAS.
